# The Combined Intervention of Aqua Exercise and Burdock Extract Synergistically Improved Arterial Stiffness: A Randomized, Double-Blind, Controlled Trial

**DOI:** 10.3390/metabo12100970

**Published:** 2022-10-13

**Authors:** Min-Seong Ha, Jae-Hoon Lee, Woo-Min Jeong, Hyun Ryun Kim, Woo Hyeon Son

**Affiliations:** 1Department of Sports Culture, College of the Arts, Dongguk University-Seoul, 30 Pildong-ro 1-gil, Jung-gu, Seoul 04620, Korea; 2Department of Sports Science, University of Seoul, 163 Seoulsiripdae-ro, Dongdaemun-gu, Seoul 02504, Korea; 3Wellcare Korea Co. Ltd., 26 Wadong-ro, Danwon-gu, Ansan-si 15265, Korea; 4Department of Physical Education, Woosuk University, 443 Samnye-ro, Samnye-eup, Wanju-gun 55338, Korea; 5Institute of Convergence Bio-Health, Dong-A University, 26 Daesingongwon-ro, Seo-gu, Busan 49201, Korea

**Keywords:** aqua exercise, burdock extract, metabolic syndrome, cardiovascular disease, vascular endothelial function

## Abstract

Metabolic syndrome (MS), characterized by the presence of risk factors for various metabolic disorders, including impaired glucose tolerance, dyslipidemia, hypertension, and insulin resistance, has a high incidence in the Asian population. Among the various approaches used for improving MS, the combination of exercise and nutrition is of increasing importance. In this randomized controlled trial, we evaluated the effects of combined aqua exercise and burdock extract intake on blood pressure, insulin resistance, arterial stiffness, and vascular regulation factors in older women with MS. A total of 42 participants were randomly assigned into one of four groups (control, exercise, burdock, and exercise + burdock) and underwent a 16-week double-blinded intervention. Blood pressure, insulin resistance, arterial stiffness, and vascular regulation factors were evaluated before and after the intervention. The 16-week intervention of aqua exercise decreased the levels of insulin, glucose, homeostasis model assessment of insulin resistance, and thromboxane A2, but increased the levels of the quantitative insulin sensitivity check index and prostaglandin I2. The combined burdock extract intake and aqua exercise intervention had an additional effect, improving the augmentation index, augmentation index at 75 beats per min, and pulse wave velocity. In conclusion, aqua exercise could improve insulin resistance and vascular regulation factors in older women with MS. Furthermore, combined treatment with burdock extract intake could improve arterial stiffness via a synergistic effect.

## 1. Introduction

Metabolic syndrome (MS) is characterized by the presence of risk factors for various metabolic disorders, including impaired glucose tolerance, dyslipidemia, hypertension, and insulin resistance [1]. According to the criteria of the National Cholesterol Education Program—Adult Treatment Panel III, MS is defined based on the presence of three or more of the following five risk indicators: abdominal obesity, hyperglycemia, hypertension, reduced levels of high-density lipoprotein (HDL) cholesterol, and increased levels of triglycerides (TG) [2]. Compared with that in Caucasians and African-Americans, the incidence of MS is high among Asians [3].

With insulin resistance and atherosclerosis as key factors, MS accelerates the increase in the risk of cardiovascular disease (CVD) [4]. Moreover, complications often arise in relation to diabetes and reduced vascular function [4]. In this regard, the incidence of CVD is correlated with the incidence of insulin resistance or hyperinsulinemia through sympathetic activation and parasympathetic inhibition [5].

Furthermore, individuals with MS have impaired endothelial function [6]. Indeed, the incidence of atherosclerosis is increased by MS, which was shown to reduce the ability of vascular endothelial cells to regulate the secretion of various compounds that promote healthy blood flow, including vasodilators—nitric oxide (NO) and prostacyclin (prostaglandin I2; PGI2)—and vasoconstrictors—endothelin-1, angiotensin II, and thromboxane A2 (TXA2) [7,8,9,10,11].

According to the obesity, hypertension, and dyslipidemia criteria, MS results from increased arterial stiffness due to carotid artery hypertrophy and increased central arterial pressure. Accordingly, individuals with MS have increased pulse wave velocity (PWV) [12,13]. In addition, CVD-related mortality is three-fold higher among individuals with MS than among healthy individuals [14,15]. Furthermore, CVD and MS risks are higher in women than in men [3,16], and older women are more susceptible to MS due to an increase in adipose tissue and insulin resistance caused by hormonal imbalances after menopause [17]. Thus, it is essential to reduce the risk of CVD in this population through effective interventions that will achieve improvement in insulin resistance and vascular function.

Exercise is a non-invasive method for increasing the resistance to oxidative stress because it increases the activity of oxidation and antioxidation damage repair enzymes through an increase in reactive oxygen species levels [18]. Exercise is well-known as an effective treatment for MS and CVD [18]. Notably, aqua exercise is an ideal form of exercise for overweight and older women, as it is a systemic exercise in which the buoyancy of water decreases the body weight by 90%, thus minimizing the risk of injury [19,20,21]. In previous studies on older women, aqua exercise was shown to effectively improve blood pressure, insulin resistance, and CVD risk factors [22,23]. Nonetheless, few studies have investigated the effectiveness of aqua exercise in older women with MS.

Recently, as a combined treatment approach involving exercise and nutrition, studies have actively investigated natural products with converged efforts. Burdock is a plant species from the Asteraceae family, which has long been used as food in East Asian regions, including Korea, China, and Japan [20,21,24,25]. Burdock has an 80% water content and is enriched with carbohydrates and dietary fibers. It is considered a healthy food with diverse bioactive compounds and was reported to improve diabetes as well as CVD. Moreover, these compounds can prevent oxidation by suppressing NO overproduction in inflammation and CVD [24,26]. Based on these previous findings, we speculated that combined treatment with aquatic exercise and burdock intake would have a more positive effect than each intervention alone in older women.

Therefore, the aim of this study was to assess the effects of combined aqua exercise and burdock intake on blood pressure, insulin resistance, arterial stiffness, and vascular regulation factors in older women with MS. We hypothesized that a 16-week combined intervention in this population would improve blood pressure and insulin resistance and induce beneficial changes in the level of arterial stiffness through improvement in vascular regulation factors.

## 2. Materials and Methods

### 2.1. Participants

This study was conducted according to the guidelines laid down in the Declaration of Helsinki and all procedures involving human participants were approved by the Institutional Review Board of Dongguk University (DUIRB-202009-07). This trial was retrospectively registered in the Clinical Research Information Service (CRIS) (Republic of Korea, KCT0007627). After explaining the study’s purpose and contents, written informed consent was obtained from all participants. 

The study population included older women aged 70−80 years, who were selected according to the Korea Adult MS Guidelines [27]. The inclusion criteria were as follows: (1) waist circumference ≥ 85 cm; (2) blood pressure ≥ 130/85 mmHg or taking antihypertensive medication; (3) fasting blood glucose level ≥ 100 mg/dL or taking antidiabetic medication; (4) TG level ≥ 150 mg/dL; and (5) HDL cholesterol level < 50 mg/dL. Individuals satisfying three or more of the above five criteria were selected. 

### 2.2. Study Design

The study was designed as a randomized, double-blind, controlled trial. To determine the effects of a 16-week intervention of burdock intake and aqua exercise, the participants were randomly divided into four groups: placebo control (CON), exercise + placebo (EX), burdock (BD), and exercise + burdock (EXBD). The CON and EX groups took placebo beverages in the same manner in which the BD and EXBD groups took the investigational beverage. The color and odor of the burdock extract and placebo beverage were similar and their containers were identical. Unblinded personnel, who were not involved in any study assessments, labeled the investigational beverage. Investigators, other site personnel, and the participants were blinded to the beverage. The total daily required beverage intake was 300 mL; the participants drank 100 mL of burdock extract or placebo beverage after breakfast, lunch, and dinner. They were instructed not to take any other health supplements or drugs. All measurements were taken twice, before and after the intervention. The study design is presented in Figure 1.

### 2.3. Burdock Extract Preparation and Composition

The burdock extract was prepared according to a method previously described by our research group [20,21,25]. In brief, after harvest in the Sangcheong region (Gyeongnam, Korea), the burdock root was washed and dried, subsequently heated at 100 °C for 3 h, and extracted at 0.7 kg/cm^2^ pressure. The main ingredients of the extract were water (98.02% ± 0.02%), crude ash (0.10% ± 0.00%), crude fat (1.12% ± 0.00%), crude protein (0.20% ± 0.00%), crude fiber (0.03%), calcium (0.004% ± 0.00%), and phosphorus (0.009% ± 0.00%) (Pukyong National University Feed & Foods Nutrition Research Center, Busan, Korea). The burdock extract was placed in small, sealed plastic containers of 100 mL, which were provided to the participants for intake. The composition of the burdock extract is summarized in Appendix A.

### 2.4. Aqua Exercise Program

The aqua exercise program used in this study was developed by revising and complementing the program designed by our research group [20]. The temperature of the swimming pool was maintained at 26–28 °C. The aqua exercise program consisted of 50 min exercise sessions performed three times per week for 16 weeks. Each session included a 5 min warm-up exercise, 40 min of main exercises, and a 5 min cool-down exercise. The exercise intensity was established in the manner conducted in our study group based on the Rating of Perceived Exertion (RPE) scores and the Polar system (RS400sd; model APAC, 90026360; Polar, New York, NY, USA), as follows: W1–5: RPE 9–10 (30–40% heart rate reserve [HRR]), W6–10: RPE 11–12 (40–50% HRR), and W11–16: RPE 13–14 (50–60% HRR) [28]. The details of the aqua exercise program are presented in Appendix A.

### 2.5. Blood Pressure

Blood pressure was measured using a digital blood pressure monitor (Jawon Medical, Daejeon, Korea) after a 10 min rest in the supine position. Measurements were taken twice with a 3 min interval in between. The mean value from the two measurements was used for analysis. When the first and second measurements differed by ≥10 mmHg, an additional measurement was taken to obtain the mean value without significant variation.

### 2.6. Blood Sampling

All participants were instructed to fast for ≥8 h before sample collection. At 8–10 a.m., 10 mL of blood was collected from the antebrachial vein by a clinical pathologist. The blood was centrifuged at 3000 rpm for 10 min in Combi-514R (Hanil, Seoul, Korea) for further analysis. All blood analyses were performed according to the procedures described by our research group [29].

#### 2.6.1. Glucose

Glucose levels were measured in serum samples. After marking the sample, reference, and blank, 20 μL of plasma and 20 μL of standard reagent were added to the sample and reference, respectively, with the addition of 3.0 mL of coloring agent. The mixture was then incubated in a 37 °C water bath. Absorbance was measured at 505 nm.

#### 2.6.2. Insulin

Insulin levels were measured in serum samples. After centrifugation, 200 μL of the supernatant was transferred to a test tube coated with anti-insulin antibody. After addition of 1.0 mL of insulin (DPC, Los Angeles, CA, USA), the mixture was incubated at 24 °C for 20 h, followed by aspiration and chemiluminescence immunoassay using an automated immunoanalyzer.

#### 2.6.3. Homeostasis Model Assessment of Insulin Resistance and Quantitative Insulin Sensitivity Check Index

The homeostasis model assessment of insulin resistance (HOMA-IR) and quantitative insulin sensitivity check index (QUICKI) are widely used as simpler and less invasive methods to evaluate insulin resistance based on fasting serum insulin and glucose levels. In this study, the below formulae reported by Matthews et al. and Katz et al. were used to calculate the HOMA-IR [30] and QUICKI [31], respectively.
HOMA-IR = [fasting insulin (mU/L) × fasting glucose (mg/dL)]/22.5QUICKI = 1/[log (fasting insulin) + log (fasting glucose)]

#### 2.6.4. PGI_2_

Whole blood was collected in an anticoagulant tube and added to a plate coated with a PGI_2_ reagent (Amersham, IL, USA). Next, 50 μL of detection reagent A was added, and the sample was incubated for 1 h at 37 °C. Subsequently, the plate was washed four times with 350 μL of wash buffer, and 100 μL of detection reagent B was added. After incubation at 37 °C for 30 min, the plate was again washed four times with 350 μL of wash buffer and incubated with 90 μL of substrate solution for 15–22 min at ambient temperature in the dark. When the reaction was completed, 50 μL of stop solution was added to each well, and absorbance was measured at 450 nm using Manifold-24 (Amersham, IL, USA).

#### 2.6.5. TXA_2_

For TXA_2_ measurement, 0.9 mL of blood was collected and immediately placed in a polystyrene tube. After adding 0.1 mL of 3.8% trisodium citrate, followed by 1 mL of physiological saline and collagen at 2 μL/mL, the mixture was heated for 15 min in a shaking water bath at 37 °C to stimulate the production of TXA_2_. After a 5 min centrifugation at 2000 rpm, the supernatant was collected for the quantification of thromboxane B_2_, an unstable product of TXA_2_ conversion, using a radioimmunoassay kit (Amersham, TRK780, IL, USA).

### 2.7. Arterial Stiffness

Arterial stiffness was measured using a non-invasive, tonometry-based PW detector (SphygmoCor; AtCor Medical, Sydney, Australia), according to the guidelines described in the Clinical Application of Arterial Stiffness, Task Force III [32]. PWV was measured based on the PW flow from the carotid to the brachial artery. The automated software of the device was used to record the PW on both ends of the artery, and the interdistance was measured using a tape measure. Next, the PWV formula was used to divide the distance (L) by the time variation (Δt) between the PWs recorded on both sides [33].
PWV = L/Δt

For the augmentation index (AIx), we calculated the pressure difference between the highest level of the central blood pressure and the augmentation point that arises at the PW refraction generated by the traveling wave advancing to the periphery to encounter the reflected wave returning to the periphery and divided it by the PWV [34]. In addition, the heart rate-corrected augmentation index at 75 beats per min (AIx@75) was estimated.

### 2.8. Sample Size Calculation

The sample size was calculated using G-power version 3.1 for Windows (Kiel University, Kiel, Germany). We estimated the sample size for this study as *n* = 40 based on the following conditions: effect size of 0.25 (default), significance of 0.05, and power of 0.70. Considering potential dropouts, a total of 46 participants were recruited.

### 2.9. Statistical Analysis

All data were statistically analyzed using IBM SPSS Statistics 27.0 (IBM Corp., Armonk, NY, USA) and expressed as means with standard deviations. The level of significance was set at *p* < 0.05. To determine the effects of the 16-week intervention on MS indicators (insulin resistance, vascular regulation factors, and arterial stiffness), two-way repeated measures analysis of variance (ANOVA) was performed with the treatment (CON, EX, BD, and EXBD) and time (pre-test and post-test) as independent variables. The Bonferroni test was used for post-hoc analysis. The post-treatment differences in the response of each variable were analyzed using the pre-test-post-test variation (Δ) by employing one-way ANOVA and Pearson’s correlation analysis. The effect size for the pre-test-post-test variation (Cohen’s d) was expressed as the mean variation [35]. 

## 3. Results

### 3.1. Participants’ Characteristics

Of the 46 enrolled participants, four dropped out of the study owing to personal reasons. Therefore, 42 participants completed the study and were included in the analysis. The characteristics of the participants are presented in Appendix A.

### 3.2. Blood Pressure

The effects of the 16-week intervention of burdock intake and aqua exercise on blood pressure and the relevant variations are shown in Figure 2. A significant time effect was detected for the systolic blood pressure (SBP; *p* = 0.039), but only the BD group showed a significant decrease from 157.13 ± 24.37 mmHg before the intervention to 146.18 ± 17.53 mmHg after the intervention (*p* = 0.029). There was also a significant group effect for the diastolic blood pressure (DBP; *p* = 0.05; Appendix A). In addition, one-way ANOVA showed no significant difference in ∆SBP and ∆DBP across all groups (Appendix A).

### 3.3. Insulin Resistance

The effects of the 16-week intervention on insulin resistance and the relevant variation are shown in Figure 3. Insulin levels showed a significant interaction effect (*p* = 0.003). In the within-group analysis, only the EX group showed a significant decrease from 32.77 ± 20.73 mg/dL to 30.17 ± 22.76 mg/dL (*p* = 0.002). Glucose levels showed significant differences both for the time (*p* = 0.029) and interaction effects (*p* = 0.004). In the within-group analysis, the EX and EXBD groups showed significant decreases from 120.24 ± 23.47 mg/dL to 95.48 ± 13.31 mg/dL (*p* = 0.001) and from 103.50 ± 24.19 mg/dL to 88.83 ± 10.76 mg/dL (*p* = 0.05), respectively. The post-hoc test showed significantly smaller (*p* = 0.034) values for the BD group than for the CON group. 

For the HOMA-IR, the interaction effect was significant (*p* = 0.002) and the within-group analysis showed a significant decrease only in the EX group from 10.45 ± 7.53 to 3.39 ± 1.77 (*p* = 0.001). For the QUICKI, the interaction effect was also significant (*p* = 0.002) and the within-group analysis showed a significant decrease in the CON group from 0.62 ± 0.44 to 0.21 ± 0.12 (*p* = 0.005) and a significant increase in the EX group from 0.14 ± 0.82 to 0.41 ± 0.27 (*p* = 0.046, Appendix A). A one-way ANOVA revealed that ∆insulin showed no significant difference per group (*p* = 0.002). The post-hoc test indicated higher values for the EX and EXBD groups than for the CON and BD groups (*p* < 0.05). 

Moreover, ∆glucose showed a significant between-group difference (*p* < 0.007) and the post-hoc test indicated significantly higher values for the EX group than for the CON group (*p* < 0.01). The ∆HOMA-IR also showed a significant between-group difference (*p* = 0.001), with higher values for the EX and EXBD groups than for the CON group (*p* < 0.05 and *p* < 0.01, respectively) and for the EX group than for the BD group (*p* < 0.05) in the post-hoc test. Similarly, the ∆QUICKI showed a significant difference among groups (*p* = 0.003), with higher values for the EX group than for the CON group (*p* < 0.05) and for the CON group than for the EXBD group (*p* < 0.01) in the post-hoc test (Appendix A).

### 3.4. Vascular Regulation Factors

The effects of the 16-week intervention on vascular regulation factors and the relevant variations are shown in Figure 4. For PGI_2_ levels, only the EX group showed a significant increase from 15.98 ± 2.95 pg/mL to 19.32 ± 6.43 pg/mL in the within-group analysis (*p* = 0.050). TXA_2_ levels showed a significant interaction effect (*p* = 0.002), and the within-group analysis showed a significant increase in the CON group from 22.43 ± 6.32 pg/mL to 24.73 ± 6.69 pg/mL (*p* = 0.003). Conversely, the EX and EXBD groups showed a significant decrease from 18.39 ± 5.11 pg/mL to 16.88 ± 4.48 pg/mL (*p* = 0.049) and from 22.42 ± 6.66 pg/mL to 20.56 ± 5.88 pg/mL (*p* = 0.021), respectively (Appendix A). 

The one-way ANOVA showed that no significant between-group difference was found for ∆PGI_2_, whereas ∆TXA_2_ showed a significant difference among groups (*p* = 0.002), with significantly higher values for the CON group than for the EX and EXBD groups (*p* < 0.01; Appendix A).

### 3.5. Arterial Stiffness

The effects of the 16-week intervention of burdock intake and aqua exercise on arterial stiffness and the relevant variations are shown in Figure 5. The AIx showed a significant interaction effect (*p* = 0.026), and the within-group analysis showed a significant increase in the CON group from 33.14 ± 8.09% to 42.00 ± 11.15% (*p* = 0.042) and a significant decrease in the EXBD group from 42.00 ± 11.54% to 31.50 ± 10.09% (*p* = 0.027). For the AIx@75, the within-group analysis showed a significant decrease only in the EXBD group, from 40.00 ± 9.44% to 32.33 ± 7.99% (*p* = 0.050). For the PWV, the interaction effect was significant (*p* = 0.023) and the within-group analysis showed that only the EXBD group had a significant increase from 8.94 ± 1.38 m/s to 9.93 ± 1.12 m/s (*p* = 0.015; Appendix A). 

The one-way ANOVA indicated that the ∆AIx showed a significant difference among groups (*p* = 0.026), with the post-hoc test indicating significantly higher values for the EXBD group than for the CON group (*p* < 0.05). The ∆AIx@75 showed no significant between-group differences. The ∆PWV showed a significant difference among groups (*p* = 0.023), with the post-hoc test indicating significantly higher values for the EXBD group than for the EX group (*p* < 0.05; Appendix A).

### 3.6. Correlation among Variations

Pearson’s correlation analysis for the variations in the measured variables showed a positive correlation between ∆SBP and ∆DBP (r = 0.657, *p* < 0.001) and between ∆glucose and ∆insulin (r = 0.674, *p* < 0.001). A positive correlation was found with the ∆HOMA-IR (r = 0.860, *p* < 0.001) and ∆TXA_2_ (r = 0.592, *p* < 0.01); however, there was a negative correlation with the ∆QUICKI (r = −0.564, *p* < 0.01). For ∆insulin, there was a positive correlation with the ∆HOMA-IR (r = 0.892, *p* < 0.001), ∆AIx (r = 0.465, *p* < 0.05), and ∆AIx@75 (r = 0.461, *p* < 0.05) and a negative correlation with the ∆QUICKI (r = −0.760, *p* < 0.001). For the ∆HOMA-IR, there was a negative correlation with the ∆QUICKI (r = −0.608, *p* < 0.01) and a positive correlation with the ∆TXA_2_ (r = 0.427, *p* < 0.05). The ∆QUICKI showed a negative correlation with the ∆AIx (r = −0.411, *p* < 0.05), while the ∆PWV negatively correlated with the ∆AIx (r = −0.622, *p* < 0.001) and ∆Aix@75 (r = −0.610, *p* < 0.01). Finally, a positive correlation was found between ∆AIx and ∆AIx@75 (r = 0.964, *p* < 0.001). The results of the correlation analysis are summarized in Appendix A.

## 4. Discussion

In this study, we hypothesized that the combined use of aquatic exercise and burdock intake would be more effective than each intervention separately for improving blood pressure, insulin resistance, vascular regulation factors, and arterial stiffness in older women with MS. Our results supported this hypothesis and revealed several novel observations. The 16-week intervention of aqua exercise decreased the levels of insulin, glucose, HOMA-IR, and TXA_2_, but increased the levels of QUICKI and PGI_2_. Independently, however, the burdock intake intervention did not show significant results for vascular function improvement. Remarkably, the combined use of burdock intake and aqua exercise had an additional effect to improve the AIx, AIx@75, and PWV.

Insulin resistance is the most common risk factor for MS. As it reduces the response of target cells to insulin owing to a decline in their sensitivity to insulin secretion, insulin resistance has a negative effect on the overall metabolism [36,37]. Characteristic outcomes caused by insulin resistance include metabolic disorders, such as type 2 diabetes, obesity, glucose intolerance, and dyslipidemia. Insulin resistance is also recognized as a CVD-related factor in atherosclerosis and hypertension and is characterized by endothelial dysfunction [38].

In older women, the degree of insulin resistance increases with a distinct reduction in physical activity because the hormonal imbalance after menopause results in increased adipose tissue accumulation, as well as increased fasting insulin and glucose levels [39]. Physical exercise is recommended as a solution to these problems and it is associated with minimal side effects. Physical exercise is effective in treating insulin resistance [40], with the levels of insulin resistance and physical exercise being inversely related [41]. In addition, participation in regular exercise not only improves antioxidation in the body but also facilitates glucose absorption in peripheral tissues and enhances insulin sensitivity by increasing the number of insulin receptors [42,43]. In a previous study in patients with type 2 diabetes, an 8-week intervention of aqua exercise significantly reduced insulin resistance [44].

The results of the present study showed a significant decrease in insulin, glucose, and HOMA-IR levels and a significant increase in QUICKI values in the EX group, suggesting that continuous participation in the aqua exercise had a positive effect on insulin resistance. Thus, additional studies should be conducted to further evaluate the preparation and composition of burdock extract for improving insulin resistance in older women with MS.

The risk factors for MS are closely associated with the progression of atherosclerosis, with cytokine secretion from adipocytes exerting a negative effect on insulin sensitivity, thereby resulting in endothelial dysfunction [8,45]. In turn, endothelial dysfunction increases the prevalence of atherosclerosis, while arterial tension is controlled by vasodilators, such as NO and PGI_2,_ and vasoconstrictors, such as TXA_2_ [46].

PGI_2_ is known for its powerful role in the induction of vasodilation and in preventing platelet coagulation [47,48]. Gamez-Mendez et al. [49] reported a decline in PGI_2_ levels in obese mice, suggesting a role for PGI_2_ in endothelial dysfunction. Conversely, TXA_2_, which exerts antagonistic effects to those of PGI_2_, is a powerful vasoconstrictor with additional roles in inducing platelet coagulation and various physiological responses, such as facilitated thrombosis and endothelial inflammation [50]. Interestingly, a close association was observed between increased platelet activation and coagulation and CVD complications [51].

Regular exercise induces endovascular shear stress and subsequent activation of calcium ion channels and phospholipases, leading to the release of PGI_2_ [52]. In a previous study in patients with hypertension, a 16-week exercise intervention was shown to increase the levels of PGI_2_ metabolites and decrease those of TXA_2_ metabolites [53]. In addition, in a human red blood cell count experiment with burdock extract, an anti-thrombotic effect was reported [54]. Furthermore, in a study in high-fat/cholesterol-diet rats, Lee et al. [55] identified a positive effect of burdock intake on vascular dysfunction.

The results of this study revealed a significant increase in PGI_2_ levels in the EX group and a significant decrease in TXA_2_ levels in the EX and EXBD groups, indicating a positive effect of exercise on insulin resistance, by enhancing endothelial function and reducing the risk of CVD in patients with MS. However, no significant effect was shown in the BD group, which highlights the need for further studies to investigate potential markers associated with endothelial dysfunction.

Insulin resistance and hyperglycemia are reported to increase cytokine production and oxidative stress, thereby compromising vascular endothelial function [56,57]. Endothelial dysfunction caused by damage to endothelial cells through structural or functional changes in vascular walls promotes or exacerbates atherosclerosis, an independent predictor of mortality due to coronary artery disease and CVD [58,59]. To detect atherosclerosis, the carotid–femoral PWV is used, whereas the Aix is used for evaluating systemic arterial stiffness [33,60]. Patients with MS are characterized by high PWV values [61].

The increase in shear stress during exercise increases the bioavailability of NO and activates sympathetic nerves to enhance vascular endothelial function [62]. Donley et al. [63] reported that an 8-week intervention of aerobic exercise in patients with MS was associated with a reduction in carotid–femoral PWV levels [64]. Burdock has antioxidant effects due to its caffeoylquinic acid content. This enhanced antioxidation capacity may reduce the content of free radicals and promote endothelial NO synthase expression, thereby increasing NO bioavailability [65]. In addition, Lee et al. [66] performed a principal component analysis of burdock and reported the presence of L-arginine, an NO precursor.

In this study, we found a significant decrease in the AIx and AIx@75 only in the EXBD group after the intervention. In addition, the largest variations in AIx and AIx@75 values (∆AIx and ∆AIx@75) were observed in the EXBD group. Furthermore, only the BD group exhibited a decline in SBP, although there was no difference in the other variables. These findings indicated that aqua exercise improved insulin resistance and vascular regulation factors in older women with MS, while burdock intake resulted in SBP reduction, with no confirmed effects on other vascular function-related variables. Therefore, it remains unclear whether burdock intake affects vascular function and insulin resistance indicators in older women with MS. However, it was reaffirmed that aquatic exercise can be effective for improving insulin resistance and vascular regulation factors in this population. Moreover, this study revealed that combined burdock intake and aqua exercise can reduce arterial stiffness in individuals with MS.

A potential limitation of this study is that direct markers of antioxidation and NO levels were not measured in response to burdock intake. Thus, additional studies are warranted to investigate the relationship between burdock intake and direct markers of antioxidation and NO. Additionally, there are some limitations pertaining to the generalizability of our findings. First, as the focus of the study was on older women with MS, it is difficult to generalize the results to men or women of other ages. Second, owing to the small sample size, the effect size in our study was limited to 70% of the power. Therefore, future studies should be performed on a larger number of study participants to support our findings.

## 5. Conclusions

In summary, burdock intake alone cannot be expected to have a significant effect in older women with MS. However, its combined use with aqua exercise, which is effective in decreasing insulin resistance and vascular regulation factors, can additionally improve arterial stiffness.

## Figures and Tables

**Figure 1 metabolites-12-00970-f001:**
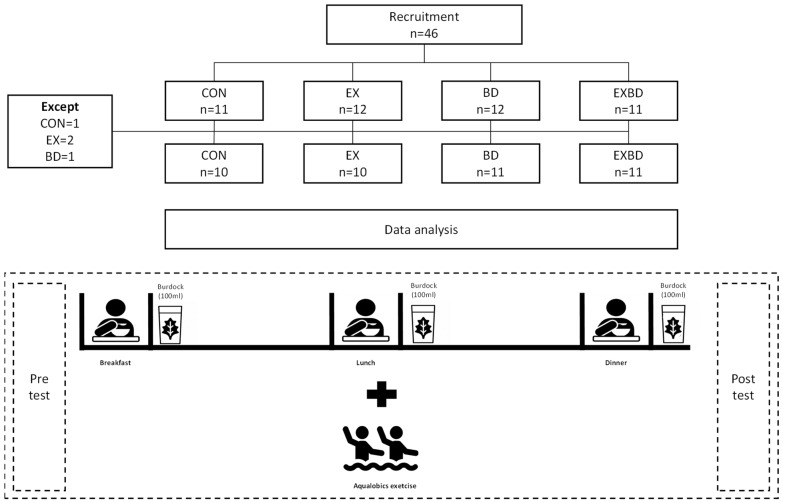
Study design. CON: placebo control, EX: exercise + placebo, BD: burdock, EXBD: exercise + burdock.

**Figure 2 metabolites-12-00970-f002:**
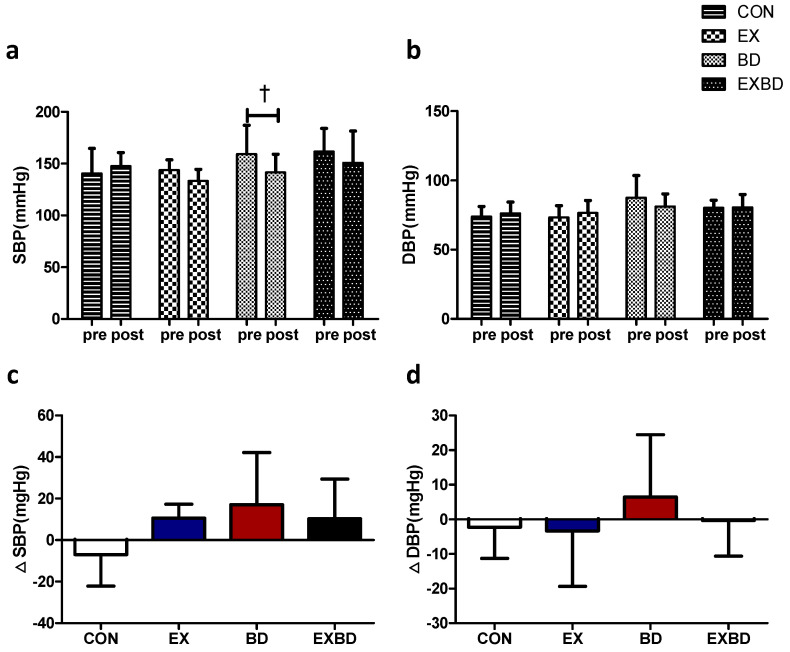
Effect of 16 weeks of aquatic aqua exercise and burdock intake on the blood pressure in older women with metabolic syndrome. (**a**) The systolic blood pressure in the BD group decreased after the 16-week intervention compared with the baseline values. (**b**) No significant changes were detected in the diastolic blood pressure in all groups. Furthermore, no significant differences in systolic (**c**) or diastolic (**d**) blood pressure variation were observed among the groups using one-way analysis of variance. Data are presented as the mean ± standard variation. ^†^
*p* < 0.05, significant; before vs. after intervention. SBP, systolic blood pressure; DBP, diastolic blood pressure; CON, placebo control, EX, exercise + placebo, BD, burdock, EXBD, exercise + burdock.

**Figure 3 metabolites-12-00970-f003:**
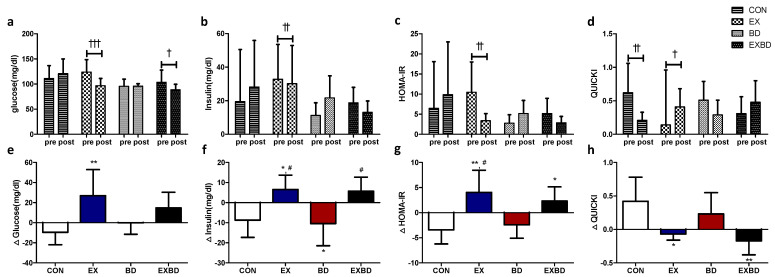
Effect of 16 weeks of aqua exercise and burdock intake on insulin resistance in older women with metabolic syndrome. After the intervention, compared with the baseline values, (**a**) glucose levels were decreased in the EX and EXBD groups, (**b**) insulin levels were decreased in the EX group, (**c**) the HOMA-IR was decreased in the EX group, and (**d**) the QUICKI was decreased in the CON group and increased in the EX group. One-way analysis of variance was used to compare the variation in insulin resistance (**e**–**h**). Data are presented as the mean ± standard deviation. ^†^
*p* < 0.01, ^††^
*p* < 0.01, ^†††^
*p* < 0.001; before vs. after intervention. * *p* < 0.05, ** *p* < 0.01 vs. CON, ^#^
*p* < 0.05 vs. BD. HOMA-IR, homeostasis model assessment of insulin resistance; QUICKI, quantitative insulin sensitivity check index; CON, placebo control, EX, exercise + placebo, BD, burdock, EXBD, exercise + burdock.

**Figure 4 metabolites-12-00970-f004:**
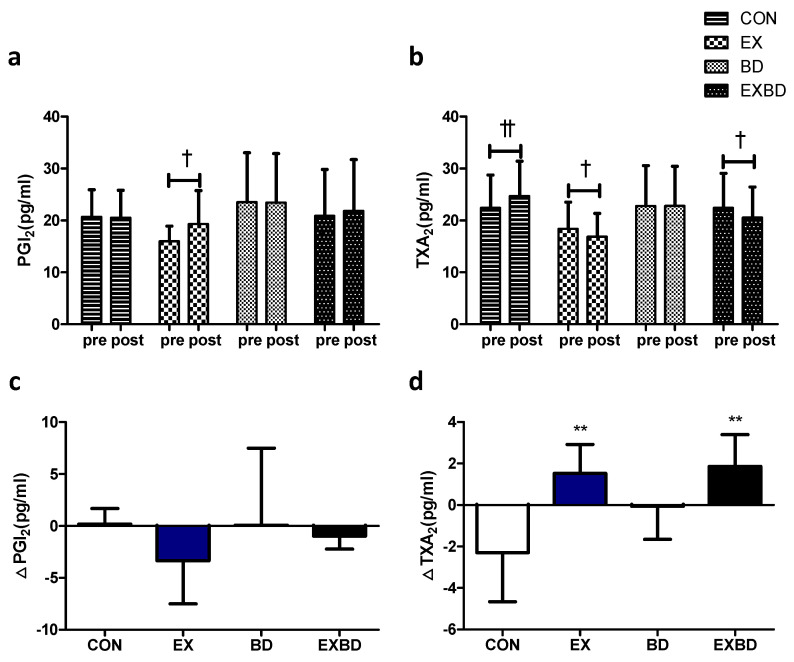
Effect of 16 weeks of aqua exercise and burdock intake on vascular regulation factors in older women with metabolic syndrome. Compared with the baseline values, (**a**) PGI_2_ levels were increased in the EX group, and (**b**) TXA_2_ levels were decreased in the CON, EX, and EXBD groups after the 16-week intervention. One-way analysis of variance was used to compare the variation in the levels of vascular regulation factors (**c**,**d**). Data are presented as the mean ± standard deviation. ^†^
*p* < 0.01, ^††^
*p* < 0.01; before vs. after intervention. ** *p* < 0.01 vs. CON. PGI_2_, prostacyclin, i.e., prostaglandin I2; TXA_2,_ thromboxane A2; CON, placebo control, EX, exercise + placebo, BD, burdock, EXBD, exercise + burdock.

**Figure 5 metabolites-12-00970-f005:**
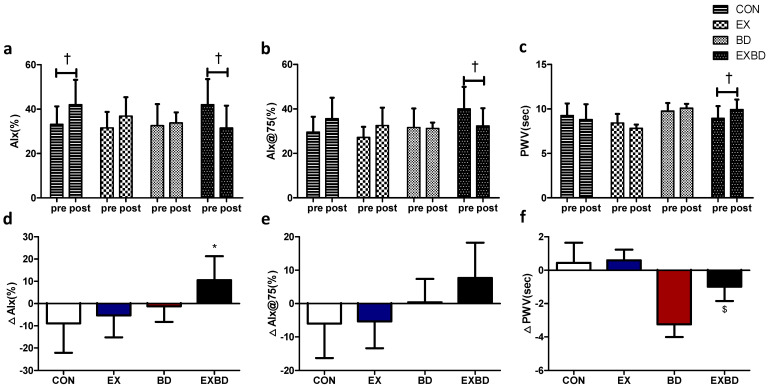
Effect of 16 weeks of aqua exercise and burdock intake on arterial stiffness in older women with metabolic syndrome. Compared with the baseline values, after the 16-week intervention, (**a**) the AIx was decreased in the EXBD group and increased in the CON group, (**b**) the AIx@75 was decreased in the EXBD group, and (**c**) the PWV was increased in the EXBD group. One-way analysis of variance was used to compare the variations in arterial stiffness (**d**–**f**). Data are presented as the mean ± standard deviation. ^†^
*p* < 0.01; before vs. after intervention. * *p* < 0.05 vs. CON, ^$^
*p* < 0.05 vs. BD. AIx, augmentation index; AIx@75, heart rate-corrected augmentation index at 75 beats per min; PWV, pulse wave velocity; CON, placebo control, EX, exercise + placebo, BD, burdock, EXBD, exercise + burdock.

## Data Availability

The authors declare that all data and materials are available to be shared upon formal request to the corresponding author.

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
