# Peer review of "The Combined Intervention of Aqua Exercise and Burdock Extract Synergistically Improved Arterial Stiffness: A Randomized, Double-Blind, Controlled Trial"

_metabolites, 2022, doi:10.3390/metabo12100970_

Round 1

Reviewer 1 Report

Thank you for improving the ms.

Two aspects will need further improvements:

The Conclusion section is a summary. The question is: What do you conclude from the results?

The references are presented twice. Please, delete one.

Author Response

Thank you for your kind and positive review. We tried to apply your comments thoughtfully. We have read your comments with great care and revised the manuscript accordingly.

We are returning herewith the above manuscript, hope that the revised manuscript will be accepted for publication in Metabolites.

Reviewer 2 Report

Major concerns include: 

1) There is no document or text that interacts with my review of this article. In the document titled, "Response to Reviewers", the authors indicate that they answer each reviewer point-by-point. However, they only interact with one of the reviewers. My review is not interacted with, so there is no evidence that the authors addressed my concerns. The text of the current version is virtually identical to the previous version providing additional evidence that the authors did not address my concerns. This is essentially the same manuscript as the previous rejected version. The only difference is that some minor English grammar changes have been made to this current version. 

2) The data have changed from the previous version to this version with absolutely no explanation for the data changes. For example, the beginning and ending values for the BP group in figure two have changed. The figures, themselves remain the same, but the values reported in the text and associated statistical analyses have changed. Multiple other data and reported statistics have changed throughout the manuscript too. The study design, parameter analyses, and statistical analyses remain the same between the two versions of the manuscript, so there should be no change in data values. 

3) One of the other reviewers of the previous version identified the fact that this study has unacceptable similarities to a previously published article by this group. There is no response to this in the current version and therefore the study remains unacceptable. 

Author Response

(The authors gave the same response as above.)

Reviewer 3 Report

Major concerns

- The whole introduction needs to be revised. The data provided are unorganized and do not create an appropriate background for the study.

- Lines 84-86: this assumption is not quite true since the authors have published other articles investigating some aspects of this topic (Ref 20,21,25). Please modify accordingly.

- Lines 113-114: It is difficult to understand how this study, which compared different interventions such as water exercise and taking an oral supplement, can be termed "double blinded." Please explain

- Lines 123-128: Who prepared and verified the composition of the extract? Were any external examinations of the composition of the extract done? Is it a commercially available product?

- Lines 219-226. Only minor BP changes have been observed after the 16 weeks of intervention. Please describe in a simpler and clearer way the results. In particular, better define the notion " only change as a function of time was observed"

- Lines 311-324: The entire table can be moved to the supplementary material

- Despite the many metabolic effects found and the indirect data of improved vascular function reported in the study no effect on blood pressure was shown. The authors are asked to explain this discrepancy in the discussion

- The discussion needs to be simplified and the data should be discussed accurately by indicating also references to tables and figures. From the data showed, interesting but not unambiguous results emerge, and the benefit provided by the association water exercise + burdock is not clearly evident in my opinion. The authors need to describe and analytically, but understandable to the reader, all the data in this view

- The conclusions need to be consistently improved

Minor issues

- The whole references are repeated twice; 71 bibliographic entries are excessive. Authors are asked to limit the number of entries to those that are truly indispensable and most scientifically relevant.

- lines 53-58: the Pubmed search "endothelial dysfunction" and "metabolic syndrome" retrieves 1311 entries. So, the assumption "Sypniewska [6] reported the presence of endothelial damage in MS patients" is not correct. Please rewrite this period.

- Lines 59-61: "Based on the criteria for obesity, hypertension, and dyslipidemia, MS is the result of hypertrophy of the carotid artery and increased central arterial pressure, leading to increased arterial stiffness". I'm not sure of the meaninig of this phrase. Please explain

- Lines 95-109: At which facility was enrollment conducted? In what period did the study take place?

- Ref 28 do not refers to "the American College of Sports Medicine (ACSM) guidelines" as reported in lines 130-131. Please correct

- Lines 377-383: Once again check references for adequacy and relevance.

Author Response

(The authors gave the same response as above.)

Round 2

Reviewer 3 Report

The authors responded quite convincingly to my suggestions.

Despite my indication, they have included among the citations of the revised version some articles that share some similarities with the current one, For example, regarding some aspects of the methodology and the part concerning PGI2 and TXA2. It would be desirable if any possible overlap could be totally eliminated

Author Response

To Reviewers:

Thank you for your kind and helpful review. We tried to apply your comments with great care and revised the manuscript accordingly. Also, we did that English language re-editing has been conducted by a native English speaker. We strongly look forward to your positive evaluation.

We are returning herewith the above manuscript, and hope that the revised manuscript will be accepted for publication in Metabolites.

Round 3

Reviewer 3 Report

This reviewer appreciates the work done by the authors, and understands their reasons for not editing the bibliographic entries as requested in previous reviews.
In light of this some additional changes are needed:
1) all references to manuscripts by the same authors who have analyzed similar aspects to those evaluated in the present study in the text should be clearly indicated as being attributable to the same research group and not to other groups.
2) Lines 356-363: These statements should be modified in accordance with the results and what is stated in the conclusions "In summary, burdock intake alone cannot be expected to have a significant effect in older women with MS. However, its combined use with aqua exercise, which is effective in decreasing insulin resistance and vascular regulation factors, can additionally improve arterial stiffness."
3) Lines 382-390: since as indicated in lines 391-397 of the present study burdock extract does not change the parameters of insulin resistance this part of the manuscript is irrelevant and can be deleted.
4) Line 394: please indicate the references whose data are not confirmed by the present study
5) Lines 433-440: since, as stated in the conclusion by the authors "burdock intake alone cannot be expected to have a significant effect in older women with MS," this part of the manuscript is not relevant and can be deleted.
6) Lines 451-452: this statement is not supported by the results and should be deleted

Author Response

Thank you for your kind and helpful review. We have revised the manuscript once again to meet your comments. We believe your comments have taken our research paper to the next level.

We strongly look forward to your positive comments.

We are returning herewith the above manuscript, and hope that the revised manuscript will be accepted for publication in Metabolites.

Round 4

Reviewer 3 Report

I thank the authors for their efforts. I have no futher concerns regarding their manuscript